# Revisiting the Raman Spectra of Carbonate Minerals

Julliana F. Alves [1], Howell G. M. Edwards [2,*], Andrey Korsakov [3] and Luiz Fernando C. de Oliveira [1,*]

[1]  NEEM—Núcleo de Espectroscopia e Estrutura Molecular, Departamento de Química, ICE, Universidade Federal de Juiz de Fora, Juiz de Fora 36036-900, MG, Brazil; julliana@ice.ufjf.br
[2]  School of Chemistry and Biosciences, Faculty of Life Sciences, University of Bradford, Bradford BD7 1DP, UK
[3]  V.S. Sobolev Institute of Geology and Mineralogy SB RAS, 630090 Nvosibirsky, Russia; korsakov@igm.nsc.ru
*  Correspondence: h.g.m.edwards@bradford.ac.uk (H.G.M.E.); luiz.oliveira@ufjf.br (L.F.C.d.O.)

**Abstract:** This work presents a new discussion about the vibrational properties of the carbonate ion displayed in several different environments. The microparameters introduced by cation substitution and different crystal lattices in addition to the crystal aggregation are present in the discussion. The work comments on how the Raman modes are affected by these changes by using data obtained with four different laser excitation sources. Raman spectra excited at 1064 nm are reported at 1 cm$^{-1}$ resolution. New observations and approaches based on the Raman modes highlight the differences observed in the relative intensity and width of the bands. The new data contribute to the understanding of these materials and their spectra, bringing new observations based on the Raman modes. This work presents a new approach highlighting the differences observed in the relative intensity and width of the Raman bands. The results indicate some evidence of the influence of the crystal habit and/or the growth of the mineral itself on the Raman spectrum. In addition, the data show the influence of cation substitution upon Raman bandwidth and the interference of the size of the spot of the laser in the measurement.

**Keywords:** carbonate minerals; calcite; aragonite; magnesite; dolomite; Raman spectroscopy

## 1. Introduction

Common carbonate minerals can be classified into four groups: the calcite group, the dolomite group, the aragonite group, and an OH-bearing group. The first two comprised minerals with a rhombohedral structure, the third is of minerals with an orthorhombic structure, and the last one comprised minerals with a monoclinic crystal structure [1]. The four carbonate groups differ not only because of their crystal structures but also by the cations present in their chemical composition [1]. Comparing the calcite and dolomite groups, the main difference between these minerals is that the first one is composed of carbonate ions associated with only one type of cation, while the second is composed of carbonate ions associated with more than one cation, maintaining the rhombohedral structure [2–5]. For these two groups, the structure is based upon alternating layers of carbonate ions and cations, comprising planes of carbonate ions parallel to planes of cations [6]. The position of the carbonate is repeated every six layers due to its triangular geometry and the 6-fold coordination of the cations [1]. For dolomite, the cation layers are alternating; that is, Ca$^{2+}$ and Mg$^{2+}$ fill alternating planes separated by planar carbonate groups [3–8].

Generally, the orthorhombic structure is favored for cations whose radius is larger than 1.1 Å in octahedral coordination, and the rhombohedral structure is favored for smaller cations [1,2,6]. Ca$^{2+}$ has an ionic radius of 1.14 Å in octahedral coordination, so it can provide minerals in both structural cases [1,6]. That is also the case for the aragonite group. When exposed to high pressure, calcium carbonate is formed like aragonite as an orthorhombic polymorph [6].

A characteristic feature of crystals is "directionality": specific directions in crystals are inherently different, and these differences are implicit in the lattice structure [1,2]. If the growth velocity was equal in all directions, crystals would occur as spheres [2]. Instead, they display a regular morphology with planar surfaces [2]. Crystal growth leads to various crystal morphologies expressed by specific combinations of crystal forms and states of aggregation that reflect the kinetic conditions [2].

The morphology and state of aggregation depends on the nucleation rate, the number of nucleation sites and the growth rate [2]. All of these are complicated functions of many parameters, including temperature, chemical composition, trace elements, and defects in the crystal structure, and in most cases, the relationships are not very well known [2]. The external appearance of a crystal, its combination of crystal forms, and the relative development of these forms are collectively called the crystal habit [1,2]. Even though the complexities of crystal growth processes lead to morphologies that are not perfect regular polyhedral, there is nevertheless a characteristic shape to many minerals, and it is used for mineral identification [2]. The same mineral may occur with an equant, acicular or fibrous habit, depending upon the conditions under which it grows [2].

Raman spectroscopy has proved to be a very powerful characterization technique for carbonate minerals [3,9–45]. For the rhombohedral groups, $CO_3^{2-}$ has five active Raman modes: three internal modes of $CO_3^{2-}$ and two external vibrations of the crystal lattice [9]. For the orthorhombic minerals (aragonite group), there are thirty Raman active fundamental phonons [13,20,21]. The vibrational modes of the carbonate ion are directly affected by all these changes in the environment: not only within each of the groups, but they are also seen as changes between the groups.

This work contains a discussion of the vibrational characterization for the carbonate ion in several different environments: not only the microparameters such as cation substitution and different crystal lattice but also the crystal aggregation. The focus of this work is the three main regions of the internal modes of the carbonate ion and the lattice modes. To compare the results using different excitation sources, the discussion is made with the obtained data at 1064 nm laser (an interferometric instrument), and 785, 632.8 and 532 nm (dispersive system). Finally, it is discussed how these data can assist in the presentation of a different approach for Raman analysis of some important carbonate minerals.

## 2. Methods

A total of 20 samples of different carbonate minerals were analyzed. Some of these samples were donated by the Centre of Mineral Technology (CETEM, Rio de Janeiro, Brazil), whereas others were purchased from local specialized mineral stores from the city of Rio de Janeiro, Brazil, and the samples were purchased and identified using the Raman technique itself. Table S1, showing all the sample information, can be seen at Supplementary material. Figure S35 presents photographs of each one of the samples, and can also be seen in Supplementary material.

Two Raman spectrometer systems were used: a Bruker FT-Raman spectrometer was used with an excitation line of 1064 nm 500 mW of laser power with 1 cm$^{-1}$ spectral resolution and 512 accumulated spectra, and a Brucker SENTERRA Raman-dispersive spectrometer was also used with 785, 632.8 and 532 nm lasers as excitation sources, 3 cm$^{-1}$ spectral resolution, 10 coadditions of 10 s of accumulation time, and 100 mW (785 nm) and 20 mW (632.8 and 532 nm) of power. All spectra were obtained at least twice for each sample to avoid thermal or photochemical damage. To ensure the reproducibility of the results, the position and intensity of each one of the Raman bands in the spectra were compared.

All samples were also submitted to analysis by energy-dispersive spectroscopy (EDS) coupled to a Hitachi 3000 tabletop scanning electronic microscope, and the results are presented in Figures S1–S29. The samples were placed directly on the sample compartment of the instrument without any pre-treatment being undertaken.

### 3. Vibrational Analysis

In the calcite group, classified as rhombohedral minerals, there are 27 vibrational modes:

$$\Gamma = A_{1g} + 2A_{1u} + 3A_{2g} + 3A_{2u} + 4E_g + 5E_u \tag{1}$$

Five of these 27 vibrational modes are Raman active, and the carbonate ion presents three main vibrational modes: one near 1400–1450 cm$^{-1}$, related to the asymmetrical stretching of the C-O bond ($E_g$ symmetry), one around 1080 cm$^{-1}$ related to the symmetrical stretching of the C-O bond ($A_{1g}$ symmetry) and one around 700–750 cm$^{-1}$ related to the COO bending mode ($E_g$ symmetry); the other two $E_g$ modes correspond to libration and translation lattice modes [9].

In the aragonite group, classified as orthorhombic minerals, there are 57 vibrational modes:

$$\Gamma = 9A_g + 6A_u + 6B_{1g} + 8B_{1u} + 9B_{2g} + 5B_{2u} + 6B_{3g} + 8B_{3u} \tag{2}$$

Thirty of these modes are Raman active, and for the internal modes of the carbonate ion, there are seven vibrational modes: one near 1450–1460 cm$^{-1}$, related to the asymmetrical stretching of the C-O bond ($B_{1g}$ symmetry), one around 1080 cm$^{-1}$ related to the symmetrical stretching of the C-O bond ($A_g$ symmetry), and three around 700–720 cm$^{-1}$ related to the COO bending mode (two with $B_{3g}$ and one $A_g$ symmetry); in addition, one mode occurs around 800 cm$^{-1}$, with $A_g$ symmetry, which is not observed at room temperature [13]. The other twenty-three modes correspond to lattice modes of the orthorhombic structure [13].

Similar to calcite, for dolomite, which is also a rhombohedral mineral, there are 18 calculated vibrational modes:

$$\Gamma = 4E_g + 4A_g + 5E_u + 5A_u \tag{3}$$

Eight of these modes are Raman active: the asymmetrical stretching of the C-O bond ($E_g$ symmetry) appears around 1440 cm$^{-1}$; the symmetrical stretching of the C-O bond ($A_g$ symmetry) appears around 1100 cm$^{-1}$, and for the COO bending mode, there are two bands with $E_g$ and $A_g$ symmetries, appearing around 720 and 880 cm$^{-1}$, respectively. All the other vibrational modes correspond to lattice movements [3,4,10,33].

The azurite (monoclinic mineral) Raman spectrum can be divided into three types of modes: the $CO_3{}^{2-}$, OH$^-$ and Cu-O modes [16,43]. Taking into account only the carbonate ions in the azurite structure, there are 24 vibrational modes:

$$\Gamma = 6A_g + 6B_g + 6A_u + 6B_u \tag{4}$$

where 12 of these modes are Raman active. For the asymmetrical stretching of the C-O bond, there are three bands, which are observed at 1416 cm$^{-1}$ ($A_g$ symmetry), 1428 cm$^{-1}$ ($B_g$ symmetry), and 1456 cm$^{-1}$ ($A_g$ symmetry). For the symmetrical stretching of the C-O bond, there is one band at 1094 cm$^{-1}$ ($A_g$ symmetry), and for the COO bending mode, there are four bands at 740 cm$^{-1}$ ($B_g$ symmetry), 763 cm$^{-1}$ ($A_g$ symmetry), 814 cm$^{-1}$ ($B_g$ symmetry) and 837 cm$^{-1}$ ($A_g$ symmetry) [16,43].

Furthermore, for malachite (a monoclinic mineral), there are 15 vibrational modes:

$$\Gamma = 3A_g + 1B_g + 1B_{2g} + 1B_{3g} + 3B_u + 3B_{2u} + 3B_{3u} \tag{5}$$

Associated with the internal modes of carbonate ion, there are six Raman active modes. For the asymmetrical stretching of the C-O bond, there are two bands at 1460 cm$^{-1}$ ($B_g$ symmetry) and 1490 cm$^{-1}$ ($A_g$ symmetry). For the symmetrical stretching of the C-O bond, there are two bands at 1066 cm$^{-1}$ ($A_g$ symmetry) and 1100 cm$^{-1}$ ($B_g$ symmetry), and for the COO bending mode, there are three bands, which can be seen at 719 cm$^{-1}$ ($A_g$ symmetry), 750 cm$^{-1}$ ($B_g$ symmetry), and 818 cm$^{-1}$ ($A_g$ symmetry) [43].

## 4. Results and Discussion

The discussion in this work will be based on the similarity of structures and crystal parameters for the investigated series of carbonate minerals. Further details will then follow on the Raman spectrum of each mineral, exploring the behavior of the structure and chemical bonds against the interaction with the excitation source under different spectral conditions.

This work reports new data for the Raman spectra of some carbonate minerals using 1064 nm excitation with a 1.0 cm$^{-1}$ spectral resolution. Figure 1 shows the data for one sample of each mineral. For siderite, azurite and malachite, it was not possible to obtain the Raman spectra for 1064 nm excitation. Siderite shows a bigger interference of thermal background, whereas azurite and malachite show photothermal decomposition, even with a very small incident laser power. For a better understanding of the related parameters, the discussion will follow considering the observations made about the Raman shift, the bandwidth, and the relative intensity.

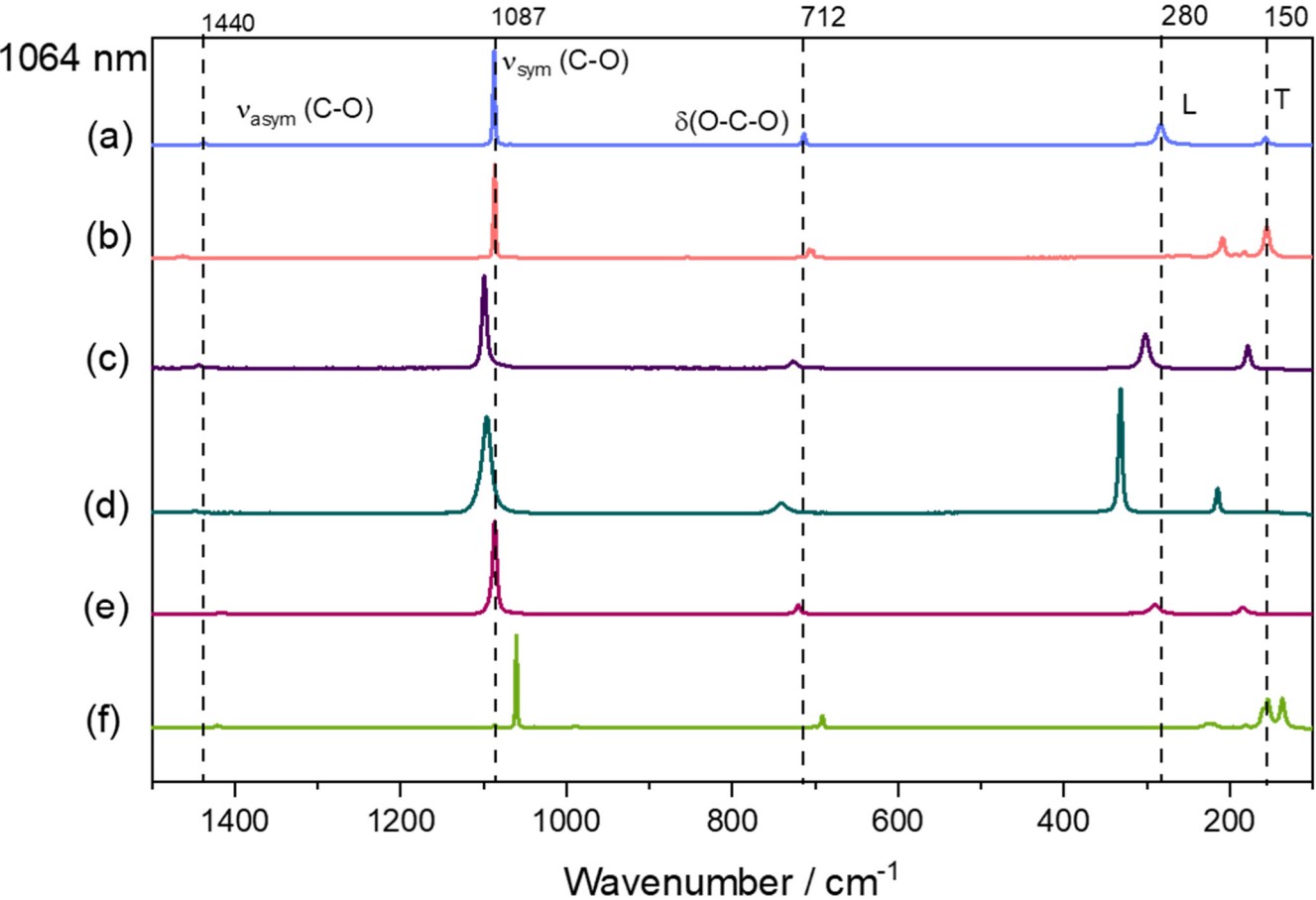

**Figure 1.** Raman spectra for one sample of each studied mineral in 1064 nm excitation source, calcite (**a**), aragonite (**b**), dolomite (**c**), magnesite (**d**), rhodochrosite (**e**) and witherite (**f**). In that figure, the spectra are not normalized.

Two parameters can be investigated when we compare samples of the same mineral and the minerals themselves: the shift of the Raman band, which is caused by the cation substitution and the intensity of the Raman bands. For the first parameter, the presence of different cations in the structure must be considered, but for the relative intensity of the bands, some other variables must also be considered. Since the relative intensity of a band is influenced by the optical orientation of the crystals, the comparison between the spectra will be made by modes from the same symmetry as reference modes. For example, in the calcite group, the asymmetrical stretching and bending modes of $CO_3^{2-}$ belong to the same

symmetry ($E_g$). Therefore, the observed differences in relative intensity were analyzed by using this as a normalized parameter, since both modes are affected in the same way by the optical orientation.

For the FT-Raman spectrometer, the system has a Ge detector, which shows the same response within the whole spectral range. Therefore, the proposed approach of comparing the relative intensities of Raman modes from the same symmetry can be used for all spectrum regions. For the dispersive instrument, the analysis must be conducted with more caution because the equipment has a CCD detector. This type of detector does not present a constant response for the entire spectral range, having more significant variations for the 785 nm excitation source.

### 4.1. Raman Shift

The first observation to be made is a shift of the vibrational mode, which is already known in the literature: for cations bigger than calcium, the Raman band shifts to a lower wavenumber, and for cations smaller than calcium, the Raman band shifts to a higher wavenumber [9]. From calcite ($CaCO_3$) to witherite ($BaCO_3$), for example, the symmetrical stretching mode shifts from 1086 to 1060 $cm^{-1}$. From calcite to magnesite ($MgCO_3$), this band shifts to 1099 $cm^{-1}$. Thus, the smaller the cation, the greater the electron density in the region, which can lead to an increase in the value of the bond strength force constant.

However, for rhodochrosite ($MnCO_3$) and siderite ($FeCO_3$), that shift pattern has not been observed. The 1086 $cm^{-1}$ Raman band is slightly shifted to 1085 $cm^{-1}$; the 1 $cm^{-1}$ spectral resolution allows that observation. The presence of the half-filled d orbital for the transition metal ions can create a different influence by their additional and concentrated negative charge, since they are cations with a smaller radius than calcium and are leading to the opposite effect [26,28,31,34,38,39,42]. Such changes were studied previously by Zhang and collaborators, showing that the substitution of calcium ions by a transition metal cation leads to differences in the vibrational properties [42], such as shifts to lower wavenumbers for the symmetrical vibrational modes.

In the dolomite spectrum, the band shift to a higher wavenumber should be less than that expected for a total replacement (which can be seen in the case of magnesite) due to a partial replacement of calcium for magnesium ions. This shift is observed for all Raman bands except for the symmetric stretching of the carbonate ion. The band is shifted to 1099 $cm^{-1}$ in dolomite and to 1093 $cm^{-1}$ in magnesite. It is important to understand that the partial cation replacement leads to a loss of symmetry in the crystal lattice unit cell as observed in the case of dolomite and predicted by the theoretical approach [3,4,10,33]. For the external modes, the shift caused by the cation substitution is the same observed in the bending mode. The decrease in symmetry in this case (when comparing dolomite, calcite and magnesite) affects each Raman mode differently, considering that the dolomite structure is the less symmetric of the three minerals. More considerations will be made for that case in Section 4.3 about the bands' relative intensity.

### 4.2. Bandwidth

The differences in the bandwidths for the vibrational modes are significant, as it can be seen at Figure 2. Calcite, aragonite and witherite present the narrowest bands, while dolomite and siderite are intermediary, and magnesite shows the widest bands. Table 1 shows bandwidth values for rhombohedral mineral samples; such values were measured based on the 1064 nm Raman spectra. It is noteworthy that the observed differences are evident when comparing different mineral carbonates, and this result is reproducible. This relationship guarantees that the samples studied do not show amorphous phases in the structure of these minerals. The presence of an amorphous phase would be evidenced by a significant broadening of the Raman band, especially in the region of lower wavenumbers; this characteristic was not observed for any of the samples here analyzed.

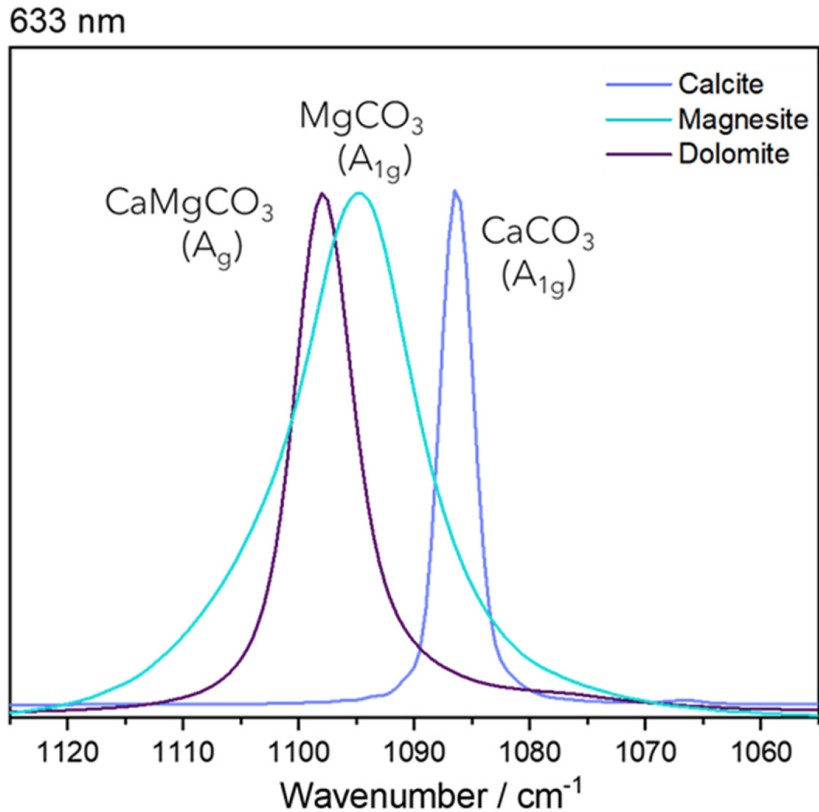

**Figure 2.** Normalized Raman spectra for the stretching mode at 633 nm excitation source for calcite, dolomite and magnesite.

**Table 1.** Bandwidth values for the rhombohedral carbonate mineral samples.

| Mineral | Bandwidth/cm$^{-1}$ |
| --- | --- |
| Calcite | 4 |
| Magnesite | 12 |
| Dolomite | 6 |
| Rhodochrosite | 6 |

Comparing the spectra for different samples, the bands show similar bandwidths for the same minerals: the difference is restricted to values close to the spectral resolution value of the equipment. So, it is possible to imply that this is a characteristic from the mineral unit cell and cannot be associated with macroscopic parameters such as the crystal aggregation in the sample or even the anisotropy. The width of a Raman band can be associated with small variations in the transition energy values associated with the vibrational modes in addition to other parameters.

Based on thermodynamics, the bandwidth can be interpreted as a consequence of the entropy; therefore, it is possible to propose that the replacement of the calcium ion by smaller cations leads to a broadening of the Raman band. This can be understood mainly for cations of similar electronic charge, such as magnesium, in the rhombohedral system. This replacement allows thermodynamics and entropic variation of the transition energy value of the vibrational mode. A combination of several different factors influences the broadening of the Raman band.

Some previous studies have shown that the positional disorder of carbonate ions creates a broadening of the Raman band, which is a parameter that can also be interpreted as a consequence of the smaller crystal size of biogenic Mg-calcite, for example [30,37]. In this specific case, the greater degree of disorder associated with a higher Mg content may be responsible for a higher chemical reactivity during diagenesis [30,37].

The proposition of this work follows the same theory. Figure 2 shows that observation, using as an example the 633 nm Raman spectra.

There are two main parameters that contribute to the intensity of a Raman band: the frequency of the excitation source and the polarizability tensor. For crystals, the incident angle of the laser source favors some symmetries, making the relative intensity of the Raman band variable depending upon the crystal optical orientation. Therefore, the crystal optical orientation is a crucial parameter in the Raman spectrum acquisition. For raw minerals, the crystal aggregation characterizes an anhedral system, so different crystals with different orientations contribute to the scattering and to the intensity and width of the Raman band. Comparing two minerals, if during the crystal growth process, there is more available space (or even on a high-pressure condition) for one of them, there will be an energy variation of the vibrational modes. This variation may be bigger for the mineral that presents less restriction, leading to an oscillation in the relative intensity of the Raman band.

Such differences could be observed by different mineral habits that are affected by temperature, pressure, concentration, etc. For example, calcite crystals grown in the presence of an excess of $Ca^{2+}$ ions exhibit an elongated habit, whereas with an excess of $CO_3{}^{2-}$ ions, the habit varies from thick to fine tabular [46]. These differences in the crystal habit for rhombohedral calcite allows us to identify how old a mineral is, based on the morphogenetic Kalb's order, that proposes the observation of some pattern for crystal growth in some cases, allowing the identification of which crystal was formed first [46].

Based on the bandwidth comparison discussion, it is important to emphasize that the observation can be made for all the samples of a same mineral, i.e., the bandwidth is the same for all the samples of a same mineral. Then, the parameter is an intrinsic property of the composition and not a consequence of the measurement conditions or minimal differences between the samples.

*4.3. Relative Intensities of Raman Bands*

For comparison purposes, as said before, the intensities of the Raman bands were normalized using same symmetry Raman modes. Calcite samples, fixing upon the asymmetrical stretching intensity, exhibit differences in the bending mode (712 cm$^{-1}$) relative intensities. In addition, all samples are raw anhedral minerals, so it is possible to disregard the crystal orientation influence upon the Raman band intensity—at least for 1064 nm excitation, which has a macroscopic configuration. In this case, the laser spot over the sample area is big enough to include diverse crystal units in several orientations, and the spectrum is an average of all the orientations over the sample spot region.

As can be seen in the Raman spectra, the C-O asymmetrical stretching mode is relatively weak and, in some cases, depending on the quality of the spectrum, it can be imperceptible for some carbonate minerals. This is the siderite case: for example, in such cases, it is possible to compare the relative intensity based on the low wavenumber region frequency, since the parameters of instrumental response previously discussed do not interfere in the analysis. For the calcite group, that approach can be used, and some small differences are observed (Figure 3).

The major difference between the samples is the crystal habit (a macroscopic visual parameter) of the mineral and its visual color. It is difficult to suggest that the differences in the relative intensity are influenced by the presence of very small quantities of different cations. The intensity of a Raman band is easily associated to the permissibility of the motion, which can be related to available space, explaining the ease of performance of the vibrational movement when the chemical species are in these particular chemical environments. It is interesting to point out that the euhedral system (sample A) does not show the higher relative intensity.

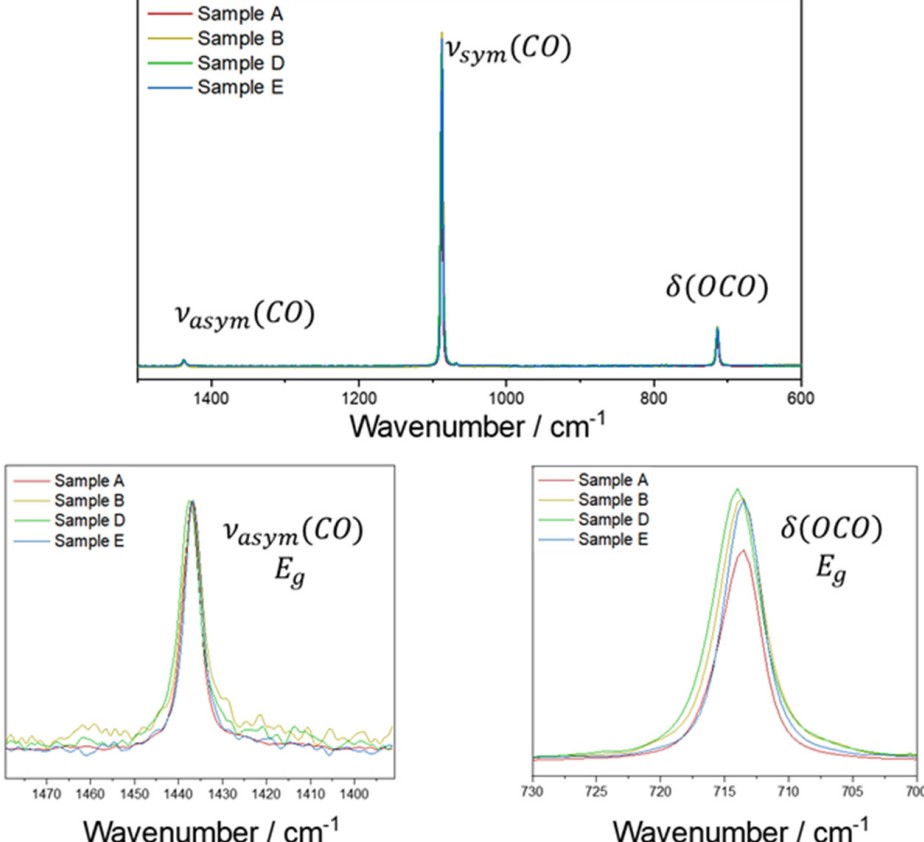

**Figure 3.** Raman at 1064 nm excitation source for calcite samples A, B, D and E. Sample C was omitted due the poor signal/noise ratio. Three regions of the spectrum are presented: the 1500–600 cm$^{-1}$ region and the two more magnified regions in 1480–1490 and 730–700 cm$^{-1}$.

The main difference observed in the SEM images for calcite samples is the changes in depth and the cleavage marks (see supplementary material). Comparing the sample with higher value of relative intensity, sample D, and the sample with the lowest, sample A, more evident marks of different depths are observed, and the straight lines of crystal growth are seen for sample A, while sample D shows smooth variations of depth and almost no straight lines of crystal growth.

These first considerations of relative intensity associated to crystal growth are very sketchy, and there is a need for more samples with defined crystal habit by a professional. Also, the analysis is limited by the number of bands that are available for comparison. In the aragonite case, these relations will be more evident.

This approach can be used comparing samples of the same mineral but also for samples of different minerals, wherein we can consider a cation substitution. Figure 4 shows a comparison between calcite, magnesite and dolomite using the C-O bending mode as a reference. Dolomite exhibits the higher relative intensity for asymmetrical stretching and similar values of T and L mode intensity; the intensity of the L mode compared to the T mode is greatest in the magnesite spectrum. The decrease in symmetry in this case (comparing dolomite, calcite and magnesite) affects each Raman mode differently, considering that the dolomite structure is the less symmetric of the three minerals. It is also the one that exhibits the highest intensity for the asymmetric stretching mode.

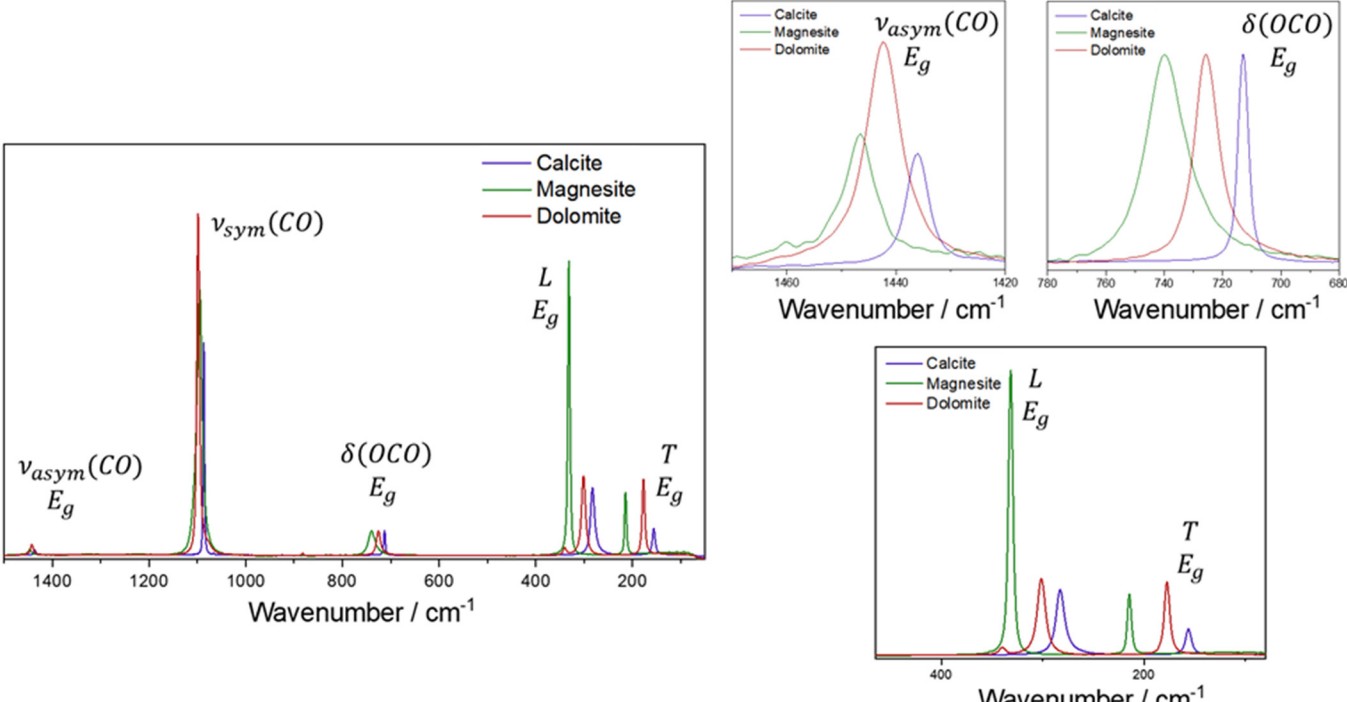

**Figure 4.** Raman spectra at 532 nm for calcite, magnesite and dolomite. Four regions of the spectrum are presented: the 1500–50 cm$^{-1}$ region, and the three more magnified regions, 1480–1420, 780–680, and 500–50 cm$^{-1}$.

Magnesite has the same symmetry as calcite, but the structure is more flexible due to the smaller cation, so we can observe wider bands. Lastly, the L mode higher intensity is expressive and can be associated with the permissibility of the movement, since these modes are correlated to collective vibrational movements.

For the aragonite samples, the symmetrical stretching mode at 1086 cm$^{-1}$ has the same symmetry as the bending mode at 705 cm$^{-1}$ (A$_g$). Hence, we can make a comparison between the relative intensities of the aragonite modes for the three samples using four different normalizations: one for each vibrational symmetry.

Figure 5 shows the spectra for all the aragonite samples at 1064 nm for each Raman mode normalized by the A$_g$ mode. Normalizing to the $\nu_{sym}$ (CO), we can see differences in the intensity of the 705 cm$^{-1}$ band. The samples do not show the presence of different cation (according to the EDS analysis, Figures S1–S29). The spectra in Figure 6 are normalized to the A$_g$ symmetry, so it is possible to compare the changes in the bands related to that symmetry: 1086, 705, 217, 164 and 145 cm$^{-1}$. The 705 cm$^{-1}$ relative intensity increases in the order C < A < B, but the A$_g$ modes in the low wavenumber region do not change at all.

Normalizing by other symmetries, for B$_{1g}$ for example, it is possible to compare the 1462 and 155 cm$^{-1}$ bands, and normalizing to the 1462 cm$^{-1}$ band, an increase in intensity for the 155 cm$^{-1}$ band can be observed. Normalizing the B$_{2g}$ (717, 208 and 182 cm$^{-1}$) bands or B$_{3g}$ symmetry (701 and 193 cm$^{-1}$ bands), no difference is observed, which is the expected behavior according to the optical orientation influence in Raman spectrum.

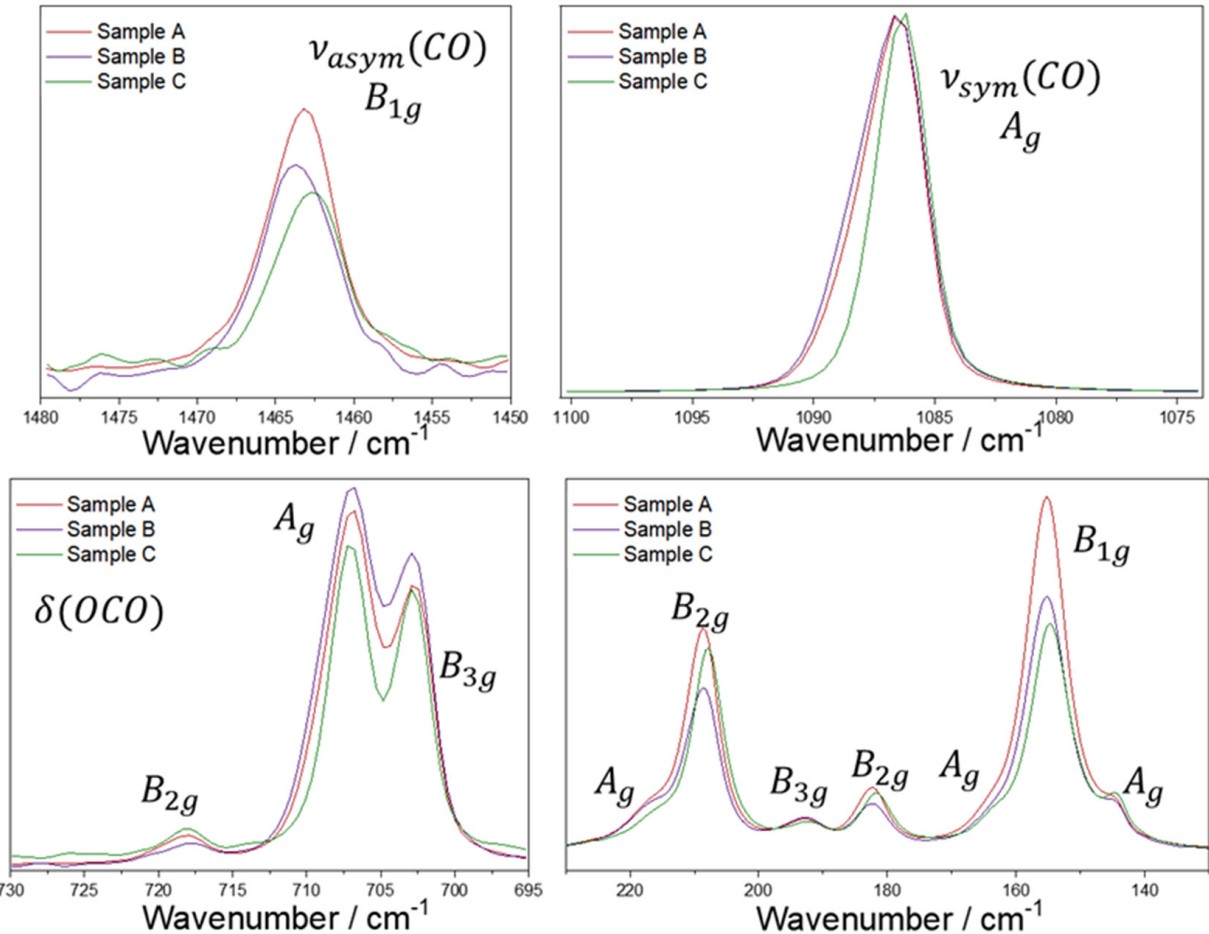

**Figure 5.** Raman spectra for aragonite samples A, B and C at 1064 nm excitation source, normalized by the $A_g$ mode at 1086 cm$^{-1}$. It presented four regions of the spectrum: 1480–1450 cm$^{-1}$, 1100–1075 cm$^{-1}$, 730–695 cm$^{-1}$, and 230–130 cm$^{-1}$. The symmetry of each mode presented was based on De La Pierre and collaborators' work [13].

The differences observed for the relative intensities are small, but we can suggest that two modes are affected: the symmetrical $A_g$ mode and the $B_{1g}$ mode. The Raman data strongly suggest the same approach that was used for the calcite samples, i.e., the relative intensity of the Raman band is being influenced by the crystallization degree of the mineral samples or, similarly, to the arrangement of the crystal growth/habit. The spectra using the $B_{1g}$, $B_{2g}$, and $B_{3g}$ modes can be seen in Figure S30 in the supplementary material.

Dolomite samples show the same Raman shift for all the carbonate modes despite their variable elemental composition, which is in agreement with the literature [9]. Comparing the relative intensities (Figure S31), the differences are largest for the external modes, mainly for the T mode. A shoulder at the symmetrical stretching mode (Figure S32) for all the samples is seen at 1078 cm$^{-1}$. An important reference work for the dolomite Raman spectra in the literature was published by Farsang and collaborators; the same shoulder is also observed for their sample, which implies that this information is intrinsic to the dolomite structure [11]. This Raman mode is not predicted theoretically, and an explanation is that it would arise from the possible appearance of different carbonate species in the solid state due to different chemical interactions with the magnesium and calcium ions in the structure. For magnesite samples (Figure S33) of the $E_g$ modes, only the L mode changes from sample A to B, but this is not a significant observed change.

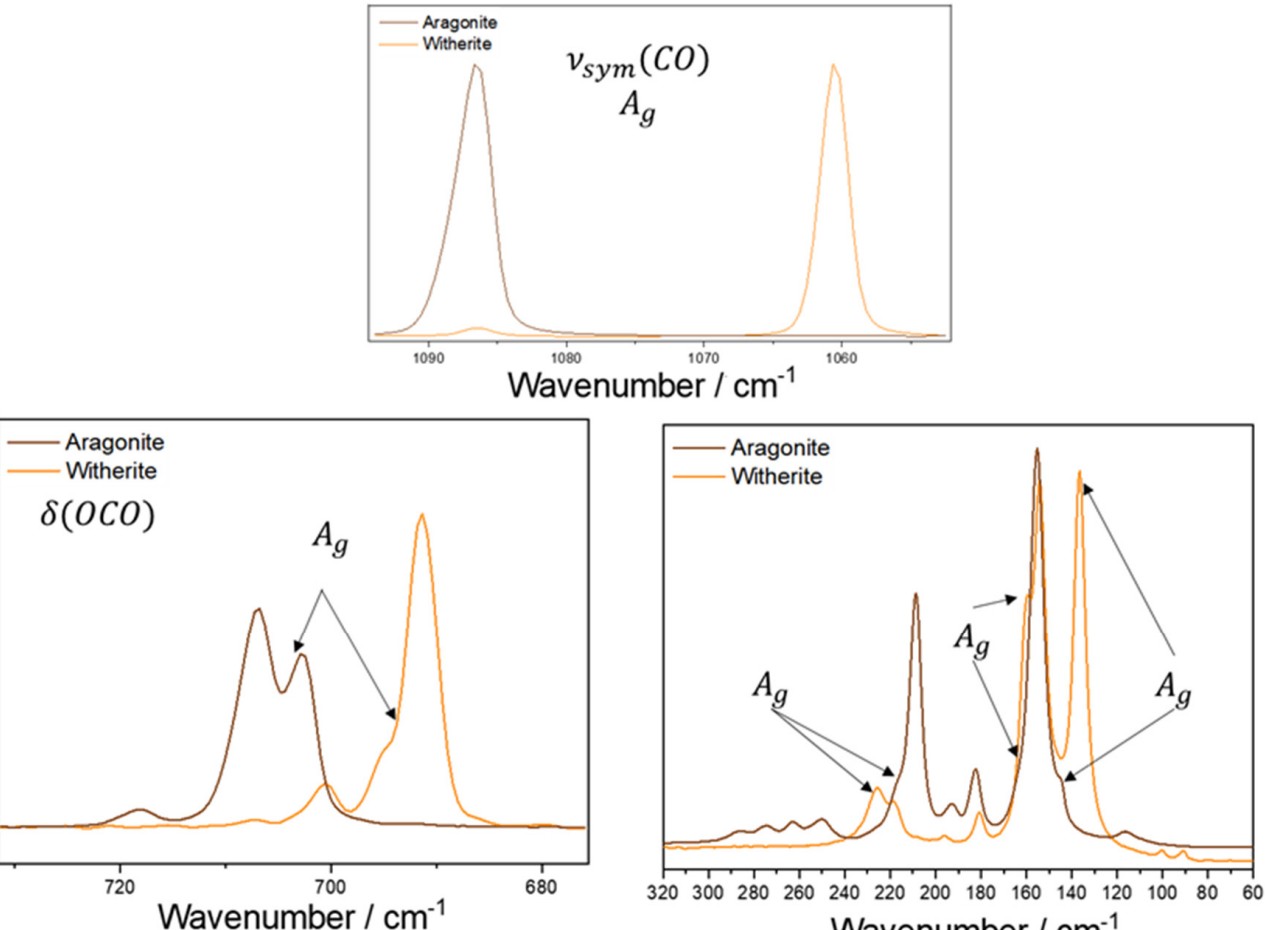

**Figure 6.** Raman spectra for aragonite and witherite samples at 1064 nm excitation source normalized by the $A_g$ mode. Three regions are presented: 1100–1050, 732–675 and 320–60 cm$^{-1}$.

Comparing aragonite and witherite, normalizing the spectra to the $B_{1g}$ symmetry mode, almost no difference is observed. For the $A_g$ symmetry, normalized by the symmetric stretching mode, the bands at 145 and 164 cm$^{-1}$ (in aragonite) are shifted to 136 and 160 cm$^{-1}$ (in witherite), respectively. We observed a higher relative intensity for these bands (136 and 160 cm$^{-1}$) in witherite, whereas the 217 cm$^{-1}$ Raman band is shifted to a high wavenumber in witherite (225 cm$^{-1}$), but the relative intensity is similar.

For the $B_{2g}$ symmetry, normalized by the bending mode (717 in aragonite and 700 cm$^{-1}$ in witherite), the relative intensity decreased in the witherite spectrum. On the other hand, the 182 cm$^{-1}$ band is shifted to a low wavenumber (180 cm$^{-1}$) in witherite, whereas the 208 cm$^{-1}$ band is shifted to a high wavenumber (218 cm$^{-1}$).

For the $B_{3g}$ symmetry, normalized by the bending mode (703 in aragonite and 690 cm$^{-1}$ in witherite), the 193 cm$^{-1}$ band in aragonite is shifted to 196 cm$^{-1}$ in witherite. A decrease in relative intensity is observed in the witherite spectrum. Figure 6 shows the spectra excited at 1064 nm for the $A_g$ mode normalization, and the other symmetries are shown in Figure S34.

The explanation follows the Idea of the cation substitution discussed previously. The $CO_3^{2-}$ internal modes are shifted to a low wavenumber due to the increase in the cation ratio. The pattern is not observed for the external modes, which also show changes in the relative intensities of the bands. The limited number of samples precludes a better understanding of the structure of witherite, but it is possible to affirm that these changes in the relative intensity are clear enough to be classified as an intrinsic characteristic of witherite.

For azurite, the Raman spectrum shows a set of Raman bands assigned to the carbonate ion asymmetric stretching mode (Figure 7). There are three bands: 1457, 1427, and 1419 cm$^{-1}$ for spectra with 633 nm excitation. For 532 nm excitation, an additional band is observed at 1492 cm$^{-1}$. Fundamental changes in the relative intensity of these bands are observed when the excitation source is changed not only between these three bands but also when they are compared to other bands in the spectrum. The data strongly suggest a resonant Raman effect is in operation, but the sample is sensitive to two of the four available excitation wavelength sources used in this work, and new and more refined data to contribute to a Raman profile are required.

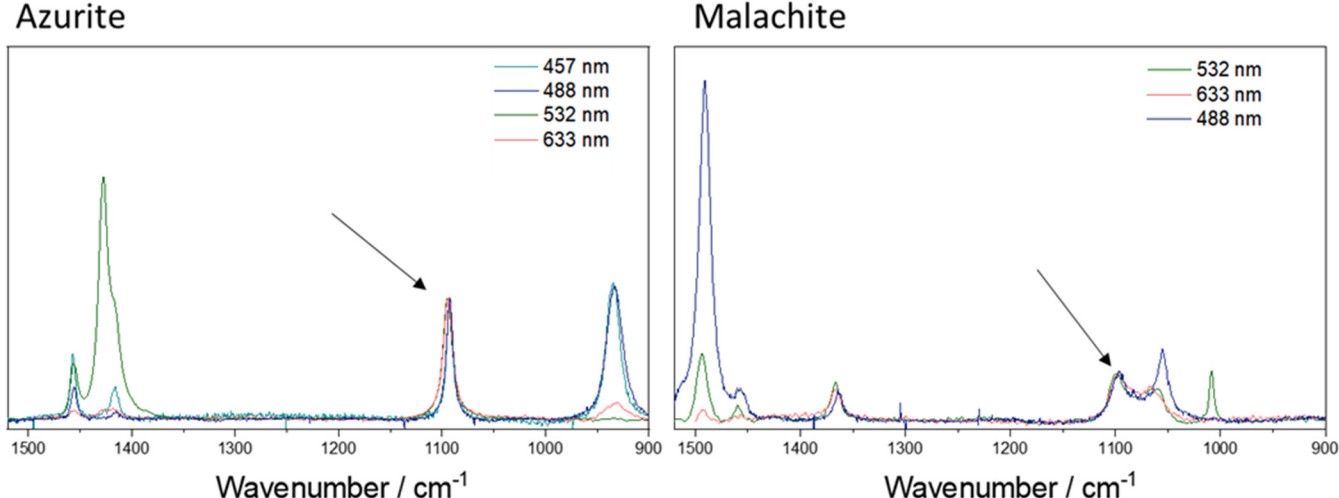

**Figure 7.** Raman spectra of azurite and malachite samples at 1520–900 cm$^{-1}$ region; the arrow indicates the symmetric stretching mode used to normalize the spectra.

Curiously, it is observed that the asymmetrical stretching became more intense than the symmetrical stretching; the reasonable explanation would be this chemical system has an electronic excited state which is degenerate, leading to a change in the polarizability according to Albrecht's approach [47]. For malachite, three bands are observed at 1364, 1460 and 1490 cm$^{-1}$ (Figure 7). With 633 nm excitation, the band at 1460 cm$^{-1}$ is very weak, but the bands at 1364 and 1490 cm$^{-1}$ have a greater intensity. For 633 nm excitation, the band at 1490 cm$^{-1}$ is more intense, and with 532 nm excitation, the band at 1364 cm$^{-1}$ is more intense. Likewise in the azurite case, changing the excitation source results in changes in the relative intensity of these bands, and the data for that mineral also suggest that a resonance Raman effect is occurring. Malachite is also a sensitive mineral to two of the four available excitation sources used here, and as stated previously, new and more refined data for a Raman profile are required.

Specifically, for malachite and azurite, two minerals showing a dependence upon the excitation radiation, the C-O symmetrical stretching mode undergoes a split due to the loss of symmetry when compared with calcite; their spectra can be seen in Figure 7. For the symmetrical stretching mode in azurite, the band shifts to a higher wavenumber (1094 cm$^{-1}$), and the symmetry ($A_g$) is maintained similar to calcite ($A_{1g}$), but the bandwidth is greater for azurite than it is for calcite. These observations are expected for the loss of symmetry (from $D_{3d}$ in calcite to $C_{2h}$ in azurite).

For malachite, Frost and collaborators described the free carbonate ions on the basis of a $D_{3h}$ point group [16]; however, Bissengaliyeva has shown in a theorical approach that the carbonate ion oxygen atoms of malachite are situated in triangular vertices where the valence angles at carbon are noticeably different from 120°, displaying a symmetry which is somewhat different from the equilateral triangle symmetry $D_{3h}$ [43]. In this last study, Bissengaliyeva has given three different force constants values for O-C-O bonds in three different environments: 0.80, 0.82 and 1.00 mdyn/Å [43]. The symmetrical stretching mode

splits, becoming three bands at 1100, 1066, and 1008 cm$^{-1}$. This last one was only seen in the Raman spectrum at 532 nm excitation.

According to Bissengaliyeva's work, the carbonate ion is not an equilateral triangle in both malachite and azurite; however, for azurite, also three values of the force constants have been evaluated for six different carbonate ions in different environments (1.40, 1.50 and 1.60 mdyn/Å) [43], yet only one band is observed in the Raman spectrum for the symmetrical stretching mode (1094 cm$^{-1}$). There is clearly some contradiction here, but the Raman spectra described in this present work are more consistent with the description of Frost, and clearly, more detailed work needs to be undertaken for these minerals [16].

For rhodochrosite, comparing with calcite (Figure 8) and using the normalization by the E$_g$ symmetry with the asymmetric stretching as a reference, it is possible to observe that the bending mode is clearly more intense, and the same observation can be made about the T mode. For the L mode, both the Raman shift and relative intensity are similar. In this case, the differences between these minerals seem to affect mainly the Raman shift, because the frequencies show a shift slightly different to that expected from the cation substitution. In terms of the relative intensity, these observations suggest that despite the cation substitution, the structure of rhodochrosite seems to behave similarly to calcite.

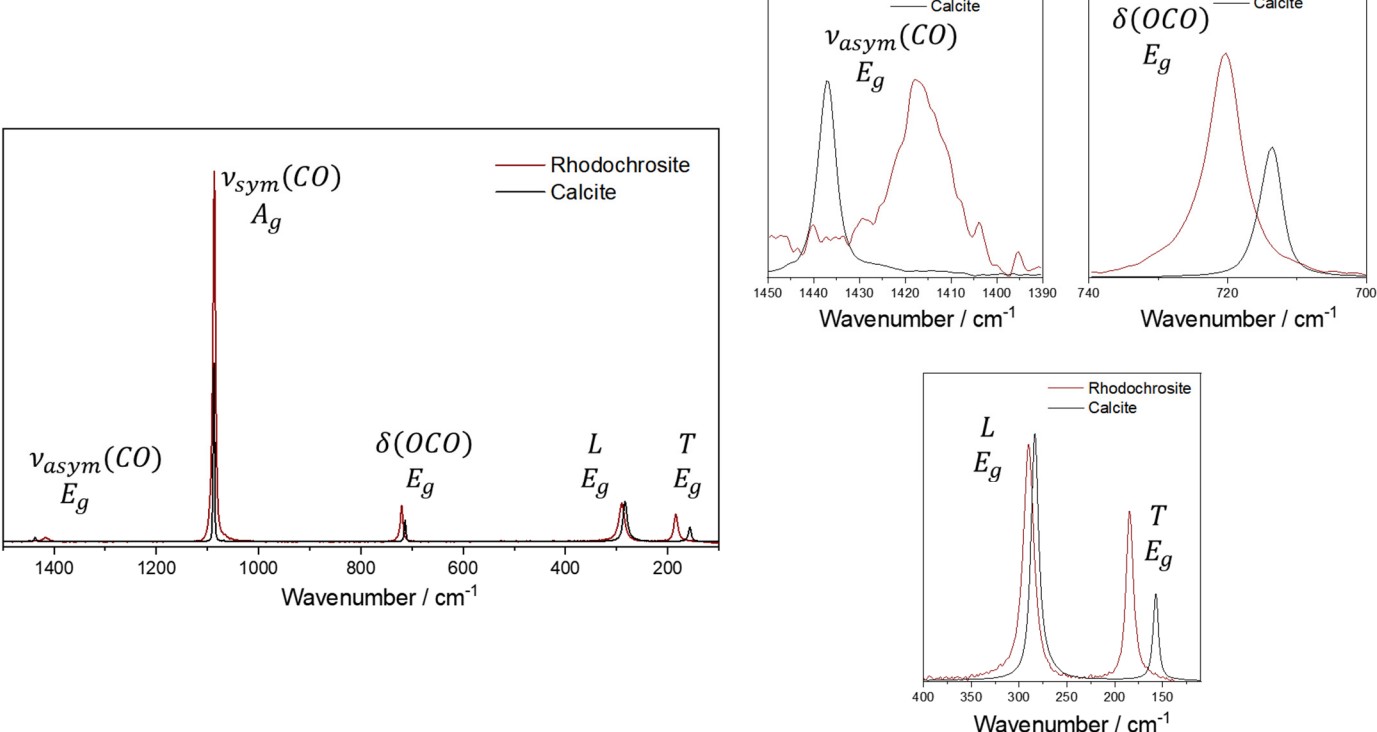

**Figure 8.** Raman spectra of calcite and rhodochrosite at 1064 nm excitation source. Four regions are presented: 1500–50 cm$^{-1}$, with magnification for 1450–1390, 740–700 and 400–120 cm$^{-1}$.

A comparison between the rhodochrosite samples was not possible due to the similarity of the two available samples; their spectra also did not show any notable difference in Raman shift and relative intensity.

For siderite, some experimental difficulties were found due to the thermal background and low Raman signal at 1064 nm. Hence, the comparison with the calcite spectrum here is made using the 523 nm excitation source (Figure 9). Using as a reference the bending mode, all the E$_g$ symmetry modes show very similar intensities except for the L mode, which exhibits a clear increase in intensity. In fact, for both rhodochrosite and siderite, the expectations are of a very similar structure; however, this is not observed. Rhodochrosite shows T and L modes that are very similar to calcite, but this is not observed in the

siderite spectrum. It is known that the siderite Raman spectrum has a strong temperature dependence and the shoulder observed in the bending mode for siderite corresponds to a vibrational mode that is shown to be infrared active in the vibrational analysis, but the flexibility of the structure of this mineral allows that this mode also becomes Raman active depending on the temperature [11,28,39,48].

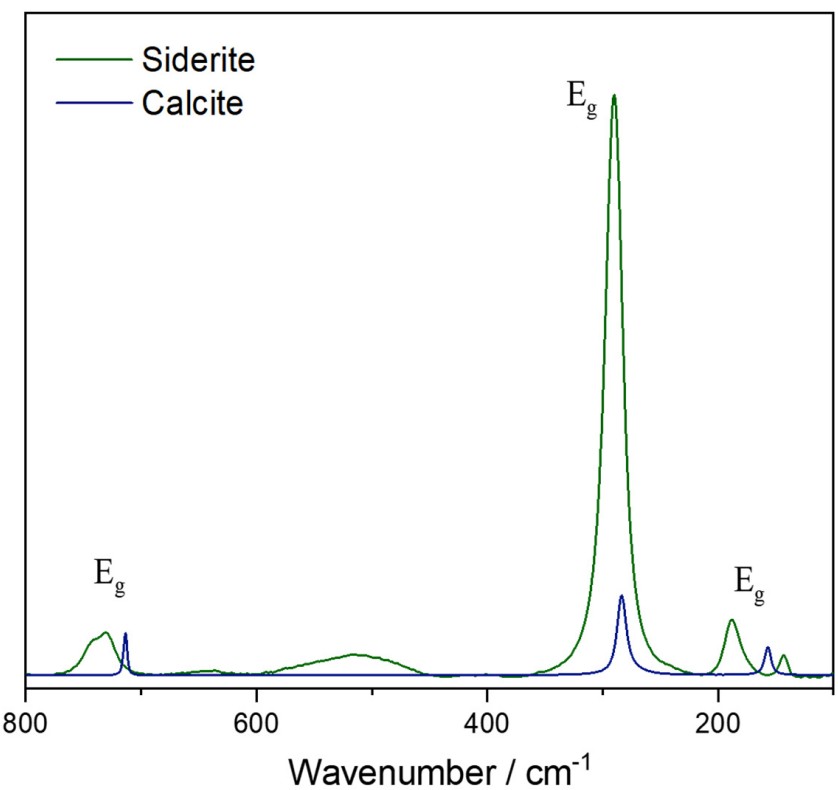

**Figure 9.** Raman spectra of siderite sample in comparison to calcite from 800 to 100 cm$^{-1}$. In this region, the bending mode next to 700 cm$^{-1}$ and the external modes can be observed.

Curiously, that dependence is not observed in the same way for rhodochrosite [11,36,39]. Data strongly suggest that these systems are completely different and unique, and the cation substitution cannot be used in the same way as in the non-transition metals case. The temperature associated with the flexibility of the crystalline lattice is noted in the literature: the small Fe atom allows distortion that leads to the appearance of an unusual mode that is only infrared allowed [11,28,39,48]. Interestingly, until this moment, siderite (FeCO$_3$) is expected to have a more flexible crystalline lattice than rhodochrosite (MnCO$_3$) because of the appearance in the Raman spectrum of the active modes in the IR. This flexibility–temperature relationship is not seen in the Raman spectrum at different temperatures for these two minerals. Rhodochrosite presents a variation of the Raman bands, both in displacement and intensity, that is much greater than siderite, that is, rhodochrosite would be more flexible if this justification were considered [11,36,39]. The contradiction exists in the literature [11,28,36,39,48] and the interaction of the carbonate anion with the transition metal is unique in each system, making these structures not comparable to each other.

### 4.4. Size of the Laser Spot

In addition to the discussion about the influence of the crystal habit, the spectra of a sample can be compared at different excitation sources. The data show that even using the same region of the sample in the measurement, changing the laser source leads to some change in the relative intensities, mainly in the low wavenumber region. This observation was made for the aragonite samples, because the same behavior was not observed for the three different samples using different excitation sources. It is possible that the laser spot

size (which is different for the wavelength of the laser excitation) changes with the average contribution of the crystal optic orientations.

Mapping the samples, it is observed that the relative intensity in the low wavenumber region changes more significantly for samples A and C but not for sample B, which implies that there is a more homogeneous average contribution of crystal orientations in this specific sample. This fact implies that this sample has more crystals organized in the same way; upon changing the laser, and consequently minimizing the spot size, the differences are clearly seen. Figure 10 shows the comparison between the spectra for aragonite B; it is important to highlight that the spectrum at 1064 nm in Figure 10 was obtained in a macroscopic configuration.

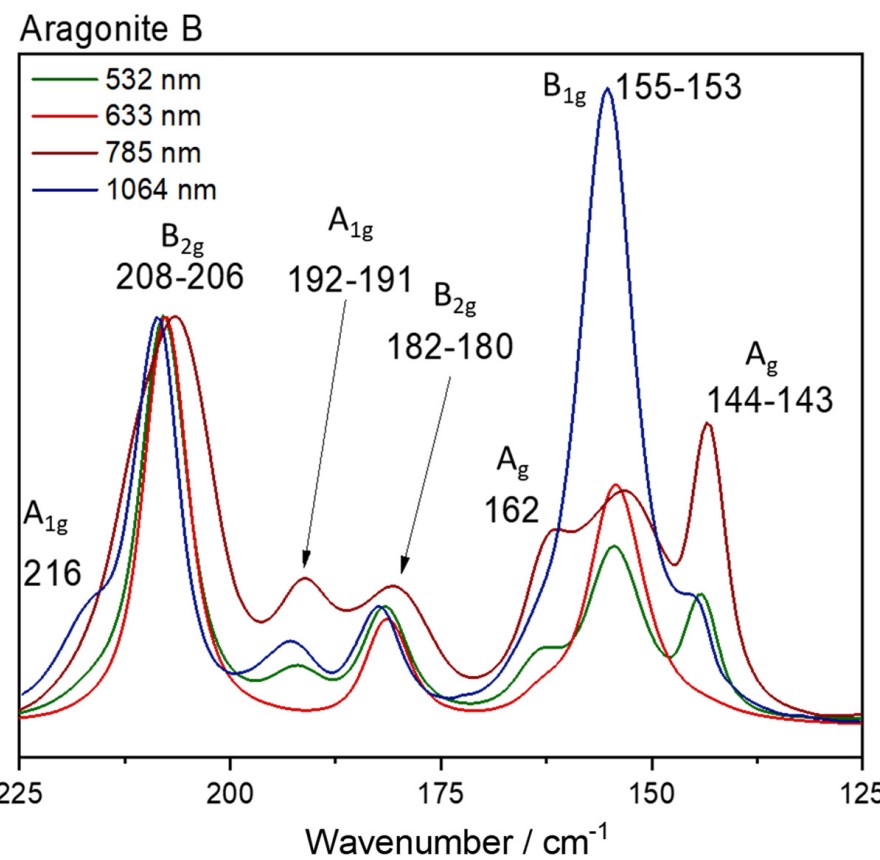

**Figure 10.** Raman spectra of one same region of aragonite sample B for different excitation sources at 225–125 cm$^{-1}$.

Changes in the external modes are also observed for rhodochrosite. For both the T and L modes presenting the same $E_g$ symmetry, the mapping of the sample shows significant changes in the relative intensity of these bands, where an inversion of the intensity can be observed depending on the region of the sample where the measurement was made (Figure 11). Initially, these differences can be related to the optical orientation of the crystals in the sample; however, both modes present the same symmetry, so they are influenced in the same way by this parameter, and an inversion of the relative intensity should not be expected.

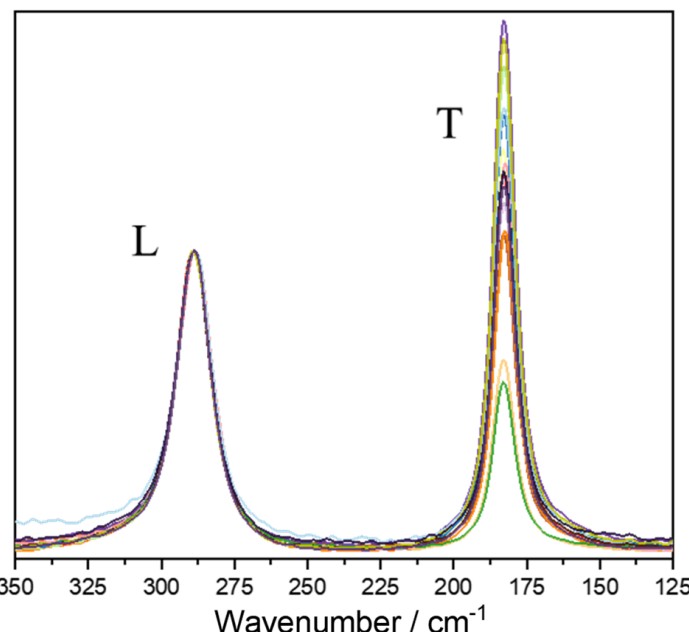

**Figure 11.** Mapping with 16 points of the rhodochrosite sample B, at 532 nm excitation source, showing the low region spectra.

## 5. Conclusions

The acquisition of Raman data excited at 1064 nm for carbonate minerals contributes to the discussion and understanding of these materials and their spectra. Perhaps the major contribution here is the indication of the influence of the crystal habit and the growth of the mineral itself to the relative intensities observed in the Raman spectrum, suggesting that different relationships between unit cells entropically affect the system, leading to changes in the relative intensities of the asymmetric stretching, bending modes of the carbonate ion and the external modes. In addition, this work has opened the discussion about the influence of the cation substitution to the Raman bandwidth, showing that small cations contribute to less rigid crystalline structures and, consequently, broader Raman bands. The identification of these minerals by their Raman spectra non-destructively demonstrates the benefits of the technique, and this revisitation of their spectra brings a proposal on how best to obtain a quality Raman spectrum: the best conditions for obtaining these spectra and how this can vary not only from one mineral to another, but from one sample to another, are explored. It is worth emphasizing that the relevance of this study of carbonate minerals for the discussion of the theory of inelastic light scattering itself shows new results associated with how thermodynamic parameters that influence the growth of a mineral, dispersed in different geological eras in the Earth's crust, can leave marks on its crystalline macrostructure that can reflect upon their vibrational spectra. In addition, the new data open the studies of how, for anhedral minerals, the same region where the measurement is obtained can lead to different Raman spectra depending on the laser wavelength due to differences in the size of the laser spot. In the case of carbonate minerals, at least for the ones investigated here, the smaller the laser spot, the better the visualization of the structural details of the sample.

**Supplementary Materials:** The following supporting information can be downloaded at https://www.mdpi.com/article/10.3390/min13111358/s1, Table S1: Samples information; Figures S1–S29: Results of EDS analysis; Figure S30: Raman spectra normalized by the $B_{1g}$, $B_{2g}$ and $B_{3g}$ symmetries for aragonite samples. Figure S31: Raman spectra normalized by the $E_g$ symmetry for dolomite samples; Figure S32: Raman spectra normalized by the $E_g$ symmetry for dolomite samples showing the bands at 1078 cm$^{-1}$ and 882 cm$^{-1}$; Figure S33: Raman spectra normalized by the $E_g$ symmetry

for magnesite samples; Figure S34: Raman spectra normalized by the $B_{1g}$, $B_{2g}$ and $B_{3g}$ symmetries for a comparison between aragonite and witherite samples; Figure S35: Samples photographs.

**Author Contributions:** Conceptualization, J.F.A. and L.F.C.d.O.; methodology, J.F.A.; software, J.F.A.; validation, J.F.A., A.K. and L.F.C.d.O.; formal analysis, J.F.A., L.F.C.d.O., H.G.M.E. and A.K.; investigation, J.F.A.; resources, L.F.C.d.O.; data curation, J.F.A. and H.G.M.E.; writing—original draft preparation, J.F.A.; writing—review and editing, L.F.C.d.O., H.G.M.E. and A.K.; visualization, J.F.A.; supervision, L.F.C.d.O.; project administration, L.F.C.d.O.; funding acquisition, L.F.C.d.O. All authors have read and agreed to the published version of the manuscript.

**Funding:** This research was funded by FAPEMIG, grant number APQAPQ-03079-23, and CNPq grant numbers 406853/2021-5 and 303569/2022-0. AK was supported by state assignment project IGM SB RAS (122041400241-5).

**Data Availability Statement:** Publicly available datasets were analyzed in this study.

**Acknowledgments:** Authors are in debt to CNPq, FAPEMIG, CAPES, FINEP (Brazilian agencies) for instrumental facilities. JFA acknowledges Petrobras for a scholarship.

**Conflicts of Interest:** The authors declare no conflict of interest.

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
