# Peer review of "Revisiting the Raman Spectra of Carbonate Minerals"

_minerals, doi:10.3390/min13111358_

Round 1
Reviewer 1 Report
Comments and Suggestions for Authors
The paper shows the results of a comprehensive study of raman spectra of carbonate minerals. This is interesting for the scientific community in general and it is worthwhile the publication in Minerals.
However, some major issues have to be addressed before publication:
- The paper is wordy and not easy to read. Two suggestions:
1. The authors should use tables to schematize the main results and make them clearer to the reader. At the moment the results are dispersed in huge amount of words. For instance, the tables could include the main considered parameters, as cation substitution, crystalline structure, crystalline habit, laser wavelength, and the main effects of these parameters on the relative intensity of the bands, the bands shift and the width.
2. The authors should separate the results in some paragraphs instead of listening all together in the same huge results and discussions paragraph. This could also help the authors to better schematize the results and provide the tables as above.
- The results are too qualitative. Some values are strongly required. For instance, instead of saying “the band is narrower than the band…” the values of the band widths are needed; a fitting of the band is the easier way for achieving the band width. Also, the relative intensity should be presented with the support of some values. I strongly suggest to list in a table the values of the band widths of the bands, at least.
- Methods paragraph: it is not clear how many measurements were carried out for each sample and for each wavelength. 512 accumulated spectra with 1064nm and for the other wavelengths? All spectra were obtained at least twice for each sample….what’s the meaning of this sentence? Please, clarify exactly this issue since it is very important from the statically point of view.
- How the authors can completely exclude the presence of a disordered or amorphous component in the samples? This would strongly influence the band width. Moreover, the broadening of the bands (like magnesite) leads to a shift of the frequency. Please add comments about this issue.
- Figures 3, 5, 6,7, 9 are made of more than one figure. In their captions a detailed description of all the included figures is needed.
- In the introduction paragraph lines 80-84 is reported that the focus of this study is around stretching and bending modes; however, in many cases comments on L and T modes are reported. Please, align this issue.
- line 283 repetition of “Sample A” is present. Please check.
- lines 270-276 very complicated sentence. Please rephrase
- line 288 “these first observations are very small” please rephrase with a clearer sentence.
Author Response
Reviewer 1:
The paper shows the results of a comprehensive study of Raman spectra of carbonate minerals. This is interesting for the scientific community in general and it is worthwhile the publication in Minerals.
However, some major issues have to be addressed before publication:
- The paper is wordy and not easy to read. Two suggestions:
- The authors should use tables to schematize the main results and make them clearer to the reader. At the moment the results are dispersed in huge amount of words. For instance, the tables could include the main considered parameters, as cation substitution, crystalline structure, crystalline habit, laser wavelength, and the main effects of these parameters on the relative intensity of the bands, the bands shift and the width.
We consider valid the proposal, but with some changes. Creating a table was taken into consideration during the construction of the manuscript, however, the data is massive. A single table would end up being too big and impractical. To help visualize numerically, we have placed a table with the bandwidth data measured from the 1064 nm spectra. we also added to Table S1 of the supplementary material a column with the crystalline structure of the studied minerals. As for the crystal habit, when the samples were donated, there was no classification regarding the crystal habit of them, but the absence of this information does not weaken the discussion of the work, because the samples are visually different. As for the displacement of the bands, this information is already widely available in the literature and the purpose of our work in revisiting the Raman spectra does not involve discussing the positions of the bands, but variations in their intensities and widths due to the replacement of the cation, crystal habit and measurement conditions.
- The authors should separate the results in some paragraphs instead of listening all together in the same huge results and discussions paragraph. This could also help the authors to better schematize the results and provide the tables as above.
We appreciate the suggestion. The text was revised and separated in topics according to the main observations and discussion. Some of the paragraphs were divided for a less dense text.
- The results are too qualitative. Some values are strongly required. For instance, instead of saying “the band is narrower than the band…” the values of the band widths are needed; a fitting of the band is the easier way for achieving the band width. Also, the relative intensity should be presented with the support of some values. I strongly suggest to list in a table the values of the band widths of the bands, at least.
We consider the suggestion valid and the values were added in the body of the text.
- Methods paragraph: it is not clear how many measurements were carried out for each sample and for each wavelength. 512 accumulated spectra with 1064nm and for the other wavelengths? All spectra were obtained at least twice for each sample….what’s the meaning of this sentence? Please, clarify exactly this issue since it is very important from the statically point of view.
Methods paragraph was rewritten to clarify the experimental part of our work. In the case of the interferometric instrument, 512 accumulated spectra were made. In the case of the dispersive instrument, 10 accumulated spectra were taken, each with 10 seconds of accumulation time. Regarding the questioned sentence, it refers to the need to obtain at least two spectra in different regions of the sample to discard the possibility of photochemical damage.
- How the authors can completely exclude the presence of a disordered or amorphous component in the samples? This would strongly influence the band width. Moreover, the broadening of the bands (like magnesite) leads to a shift of the frequency. Please add comments about this issue.
The observation is important. The samples are natural crystals. The width of the bands, even with the variation caused by cation substitution, is still characteristic of crystalline systems. The Raman bands in the low frequency region in addition to the reproducibility of the band widths for different samples of the same mineral also support a crystalline system argument. The replacement of the cation is responsible for the shift of the Raman band due the difference caused in the coulombic interaction between cation and anion in the mineral structure. This discussion is old in the literature and some articles that were references for this work are cited in the text. The proposal we make is that a smaller cation contributes to a more flexible crystalline structure, susceptible to greater energy variations in entropic parameters for the same vibrational mode, leading to a broadening of the Raman band.
- Figures 3, 5, 6, 7, 9 are made of more than one figure. In their captions a detailed description of all the included figures is needed.
The suggestion was accepted and the information was complemented.
- In the introduction paragraph lines 80-84 is reported that the focus of this study is around stretching and bending modes; however, in many cases comments on L and T modes are reported. Please, align this issue.
The suggestion was accepted and the information was corrected.
- line 283 repetition of “Sample A” is present. Please check.
The suggestion was accepted and the information was corrected.
- lines 270-276 very complicated sentence. Please rephrase
The suggestion was accepted and the writing was redone.
- line 288 “these first observations are very small” please rephrase with a clearer sentence.
The suggestion was accepted and the writing was redone.
Reviewer 2 Report
Comments and Suggestions for Authors
This study is very helpful for the deep understanding of the geological environment in which carbonate minerals form. I agree with publishing this paper in the journal.
But the manuscript has several minor requiring revision:
The content of lines 30 to 36 is redundant on page 1, should be deleted.
The signs of the vibrational modes is inconsistent, such as Eg and E?.
Author Response
Reviewer 2:
This study is very helpful for the deep understanding of the geological environment in which carbonate minerals form. I agree with publishing this paper in the journal.
But the manuscript has several minor requiring revision:
The content of lines 30 to 36 is redundant on page 1, should be deleted.
The suggestion was accepted and the writing was redone.
The signs of the vibrational modes is inconsistent, such as Eg and E?.
The suggestion was accepted and the writing was redone.
Reviewer 3 Report
Comments and Suggestions for Authors
This manuscript studies the Raman spectra of different carbonates and discusses the effects of cation substitution, different crystal lattices, and crystal aggregation on Raman intensities and bandwidth. Spectra were excited using four different laser wavelengths with the 1064 nm data being less common and compared.
I want to highlight that the usage of the Raman technique and their discussion are of good quality. Although I believe this manuscript is a solid study, giving a helpful overview how the spectral signatures are influenced by the individual factors mentioned above, I am unsure about the novelty and impact of the research. The authors claim the 1064 nm data is new, but IMHO its more unusual, but not uncommon. Reading it felt more than a review article than exciting new research. Furthermore, I would have loved to see more discussion about non-classical crystallization (e.g., the presence of amorphous phases) – something that is not mentioned at all. IMHO, the SEM analyses are not very helpful and are not adding any information. The EDS data is insufficient to precisely determine elemental ratios, as it is influenced by surface topography. Here ICP or microprobe data would be necessary to obtain data similar to what is shown in the SI.
For this reason, in my opinion the work is suited for Minerals, but requires a major revision. Especially the elemental composition should be determined by a different, more accurate method.
In addition, ALL figures need to be changed: Units shouldn't be given in brackets, instead use Wavenumber / cm-1 as stipulated by internation standards (e.g., NIST here: https://www.nist.gov/pml/special-publication-811/nist-guide-si-chapter-7-rules-and-style-conventions-expressing-values).
Furthermore, please ensure high resolution figures are being uploaded, some Figures have very low resolution.
For additional and more detailed remarks, please find all my notes in the attached files (manuscript and SI).

Comments on the Quality of English LanguageFurthermore, the English language needs to be improved – sometimes it is very hard to understand the discussion because of wrong grammar and long sentences (see attached files).
Author Response
Reviewer 3:
This manuscript studies the Raman spectra of different carbonates and discusses the effects of cation substitution, different crystal lattices, and crystal aggregation on Raman intensities and bandwidth. Spectra were excited using four different laser wavelengths with the 1064 nm data being less common and compared.
I want to highlight that the usage of the Raman technique and their discussion are of good quality. Although I believe this manuscript is a solid study, giving a helpful overview how the spectral signatures are influenced by the individual factors mentioned above, I am unsure about the novelty and impact of the research. The authors claim the 1064 nm data is new, but IMHO its more unusual, but not uncommon. Reading it felt more than a review article than exciting new research. Furthermore, I would have loved to see more discussion about non-classical crystallization (e.g., the presence of amorphous phases) – something that is not mentioned at all. IMHO, the SEM analyses are not very helpful and are not adding any information. The EDS data is insufficient to precisely determine elemental ratios, as it is influenced by surface topography. Here ICP or microprobe data would be necessary to obtain data similar to what is shown in the SI.
For this reason, in my opinion the work is suited for Minerals, but requires a major revision. Especially the elemental composition should be determined by a different, more accurate method.
We appreciate the observation and consider as relevant. At the level of comparison and interference, elementary analysis via EDS is enough to associate it with the Raman effect. In the case of Raman spectroscopy, the phenomenon is even more superficial than in the energy dispersive technique. The radiation for an EDS instrument has more penetration and occupies a larger area of the sample than the laser in the visible region for the Raman technique. Furthermore, the interference caused by the modification in the composition would be much more significant than what is pointed out in the work. The case of magnesian calcites is cited as a reference in this work, which are not dolomites, but calcites with small amounts of magnesium trapped in the crystalline structure. This property leads not only to the broadening of the band, but to its displacement. As the data did not show such behavior and the EDS did not show the presence of other metals, the possibility that the observations made were due to the elemental composition was ruled out. For this reason, we believe it is not necessary to use another technique for this work. We still reinforce the idea that EDS data are not disposable, but a small complement to work that relies exclusively on the Raman technique.
In addition, ALL figures need to be changed: Units shouldn't be given in brackets, instead use Wavenumber / cm-1 as stipulated by internation standards (e.g., NIST here: https://www.nist.gov/pml/special-publication-811/nist-guide-si-chapter-7-rules-and-style-conventions-expressing-values).
The suggestion was accepted and the correction was made.
Furthermore, please ensure high resolution figures are being uploaded, some Figures have very low resolution.
The suggestion was accepted and the correction was made
For additional and more detailed remarks, please find all my notes in the attached files (manuscript and SI).
Comments were taken into account and several corrections were applied.
A question was asked about the stores specializing in minerals mentioned in the methodology. These departments are Brazilian local stores located in the city of Rio de Janeiro, Brazil, and te samples were purchased and identified using the Raman technique itself.
It was also asked whether the laser power on the sample was measured. Perhaps the way the text was written made it confusing, but the power values presented refer to the available values and are selected through the equipment's software.
There is also a question about the treatment of the samples before being placed in the EDS compartment. No, there was no treatment, the samples are not wet, so it was not necessary.
There is a question about the use of water immersion objective. No, it was not used. All samples are solids.
I emphasize here that a scale was requested for the photographs of the mineral samples. The sample size variation is significant, but irrelevant when it comes to measuring the spectra because we used a 50X magnification objective. Therefore, the size of the samples does not contribute anything to the discussion that is restricted to observations of the Raman spectra.
Comments on the Quality of English Language
Furthermore, the English language needs to be improved – sometimes it is very hard to understand the discussion because of wrong grammar and long sentences (see attached files).
Round 2
Reviewer 1 Report
Comments and Suggestions for Authors
Good
Author Response
All references were revised in this new version of the manuscript.
Reviewer 3 Report
Comments and Suggestions for Authors
Dear authors,
thanks for adressing a lot of my questions. Unfortunately, i would have hoped to not only get answers personally, but also see some changes made in the manuscripts adressing these issues. In detail:
1) Elemental composition by EDS
We appreciate the observation and consider as relevant. At the level of comparison and interference, elementary analysis via EDS is enough to associate it with the Raman effect. In the case of Raman spectroscopy, the phenomenon is even more superficial than in the energy dispersive technique. The radiation for an EDS instrument has more penetration and occupies a larger area of the sample than the laser in the visible region for the Raman technique. Furthermore, the interference caused by the modification in the composition would be much more significant than what is pointed out in the work. The case of magnesian calcites is cited as a reference in this work, which are not dolomites, but calcites with small amounts of magnesium trapped in the crystalline structure. This property leads not only to the broadening of the band, but to its displacement. As the data did not show such behavior and the EDS did not show the presence of other metals, the possibility that the observations made were due to the elemental composition was ruled out. For this reason, we believe it is not necessary to use another technique for this work. We still reinforce the idea that EDS data are not disposable, but a small complement to work that relies exclusively on the Raman technique.
--> I agree with the authors here, but it is still scientifically incorrect to determine elemental ratios from EDS if there is surface topography - i would strongly suggest to remove Figures S21-S29, because the values are error-prone and therefor not valid. It should be enough to keep the EDS spectra to make your point.
2) Non-classical nucleation
--> I suggested to discuss aspects of non-classical nucleation, including amorphous phases - this hasn't been adressed at all.
3) The SEM pictures
I pointed out, that these images are not really helpful. The authors added a statement now (p8, line 293): 'These first observations are very sketchy and there is a need for more samples with defined crystal habit by a professional." So why not remove these images or at least put them in the SI?
4) Stores specializing in minerals mentioned in the methodology
These departments are Brazilian local stores located in the city of Rio de Janeiro, Brazil, and te samples were purchased and identified using the Raman technique itself.
--> Why not add this information to the text? If you buy chemicals you also have to give the name of the manufacturer, for me, this is the same case here.
5) There is a question about the use of water immersion objective. No, it was not used. All samples are solids.
--> Water immersion objectives can also be used on solids, as they can prevent laser-induced burning of the samples, which might help with the azurite and malachite samples.
6) Scale for the photographs of the mineral samples.
The sample size variation is significant, but irrelevant when it comes to measuring the spectra because we used a 50X magnification objective. Therefore, the size of the samples does not contribute anything to the discussion that is restricted to observations of the Raman spectra.
--> I understand that we talk about a figure in the SI, but even so, every figure - especially those of minerals - needs a scale. Especially if the size of the minerals vary so much. It is just good scientific practice to describe your samples appropriately, even if it has no effect on the Raman data.
Overall, i am not very satisfied with the revised version. It feels like all minor issues have been resolved, but the major points have remained unadressed.
Comments on the Quality of English Language6) Comments on the Quality of English Language
Furthermore, i noticed a lot of incorrect use of the English language. This still needs to be improved – especially in the corrected and added parts.
Author Response
1) Elemental composition by EDS
I would strongly suggest removing Figures S21-S29, because the values are error-prone and therefore not valid. It should be enough to keep the EDS spectra to make your point.
OK, we have done it, by removing Figures S21-S29 and maintaining only figures of SEM images and EDS spectra.
2) Non-classical nucleation
--> I suggested to discuss aspects of non-classical nucleation, including amorphous phases - this hasn't been addressed at all.
To clarify this issue, we added the following text in item 4.2: “Table 1 shows band width values for rhombohedral mineral samples; such values were measured based on the 1064 nm Raman spectra. It is noteworthy that the observed differences are evident when comparing different mineral carbonates, and this result is reproducible. This relationship guarantees that the samples studied do not show amorphous phases in the structure of these minerals. The presence of an amorphous phase would be evidenced by a significant broadening of the Raman band, especially in the region of lower wavenumbers; this characteristic was not observed for any of the samples here analyzed”
3) The SEM pictures
I pointed out, that these images are not really helpful. The authors added a statement now (p8, line 293): 'These first observations are very sketchy and there is a need for more samples with defined crystal habit by a professional." So why not remove these images or at least put them in the SI?
OK, we have done it, the image was removed.
4) Stores specializing in minerals mentioned in the methodology
--> Why not add this information to the text? If you buy chemicals you also have to give the name of the manufacturer, for me, this is the same case here.
- It was done, the information was added.
6) Scale for the photographs of the mineral samples.
--> I understand that we talk about a figure in the SI, but even so, every figure - especially those of minerals - needs a scale. Especially if the size of the minerals vary so much. It is just good scientific practice to describe your samples appropriately, even if it has no effect on the Raman data.
- It was done, we added new photographs of the samples in supplementary material, now with scale.
6) Comments on the Quality of English Language
We review the text, and several corrections were made in the language. Phrases had been rewritten and highlighted in blue color. We have done our best correcting the language.
Round 3
Reviewer 3 Report
Comments and Suggestions for Authors
We gratefully thank the authors for adressing all the comments made in the 2nd round. The raman data and interpretation is very good and will definitely be helpful for interested readers.